# Construction and Stability of All-in-One Adenovirus Vectors Simultaneously Expressing Four and Eight Multiplex Guide RNAs and Cas9 Nickase

**DOI:** 10.3390/ijms25168783

**Published:** 2024-08-12

**Authors:** Tomomi Nakahara, Hirotaka Tabata, Yuya Kato, Ryoko Fuse, Mariko Nakamura, Megumi Yamaji, Nobutaka Hattori, Tohru Kiyono, Izumu Saito, Tomoko Nakanishi

**Affiliations:** 1Department of Immune Medicine, National Cancer Center Research Institute, Tokyo 104-0045, Japan; tonakaha@ncc.go.jp; 2Laboratory of Virology, Institute of Microbial Chemistry (BIKAKEN), Microbial Chemistry Foundation, Shinagawa-ku, Tokyo 141-0021, Japan; 3Department of Pharmaceutical Engineering, Faculty of Engineering, Toyama Prefectural University, Toyama 939-0398, Japan; 4Center for Biomedical Research Resources, Juntendo University Graduate School of Medicine, Bunkyo-ku, Tokyo 113-8421, Japan; m.nakamura.ou@juntendo.ac.jp (M.N.); m.yamaji.bj@juntendo.ac.jp (M.Y.); nakanishi-t@juntendo.ac.jp (T.N.); 5Department of Neurology, Juntendo University Graduate School of Medicine, Tokyo 113-8421, Japan; nhattori@juntendo.ac.jp; 6Exploratory Oncology Research and Clinical Trial Center, National Cancer Center, Chiba 277-8577, Japan; tkiyono@east.ncc.go.jp; 7Department of Physiology, Juntendo University Graduate School of Medicine, Bunkyo-ku, Tokyo 113-8421, Japan

**Keywords:** CRISPR/Cas9, genome editing, adenovirus vector, all-in-one, multiplex guide RNA, shortened U6 promoter, gene therapy

## Abstract

CRISPR/Cas9 technology is expected to offer novel genome editing-related therapies for various diseases. We previously showed that an adenovirus vector (AdV) possessing eight expression units of multiplex guide RNAs (gRNAs) was obtained with no deletion of these units. Here, we attempted to construct “all-in-one” AdVs possessing expression units of four and eight gRNAs with Cas9 nickase, although we expected obstacles to obtain complete all-in-one AdVs. The first expected obstacle was that extremely high copies of viral genomes during replication may cause severe off-target cleavages of host cells and induce homologous recombination. However, surprisingly, four units in the all-in-one AdV genome were maintained completely intact. Second, for the all-in-one AdV containing eight gRNA units, we enlarged the E3 deletion in the vector backbone and shortened the U6 promoter of the gRNA expression units to shorten the AdV genome within the adenovirus packaging limits. The final size of the all-in-one AdV genome containing eight gRNA units still slightly exceeded the reported upper limit. Nevertheless, approximately one-third of the eight units remained intact, even upon preparation for in vivo experiments. Third, the genome editing efficiency unexpectedly decreased upon enlarging the E3 deletion. Our results suggested that complete all-in-one AdVs containing four gRNA units could be obtained if the problem of the low genome editing efficiency is solved, and those containing even eight gRNA units could be obtained if the obstacle of the vector size is also removed.

## 1. Introduction

CRISPR/Cas9 technology is expected to offer novel therapies of genome editing for various diseases [1,2,3,4,5,6], including cancer [7,8,9,10], inherited diseases [11,12,13,14], and infectious diseases [15,16,17]. Using this technique, viral vectors are expected to become effective curative agents for patients for whom effective therapies are not currently available [18,19].

E1- and E3-deleted adenovirus vector (AdV), also called replication-incompetent AdV, is widely used both in basic studies and in applied fields including gene therapy [18,19]. AdVs show high level expression, infect a wide variety of cells and tissues, and transfer an expression unit to both dividing and non-dividing cells. Their large-scale production can also be achieved. These features are advantageous for practical application for promising drugs. Furthermore, AdVs possess a large capacity, up to approximately 8 kilobases (kb) in length. However, a major drawback of AdVs is their high immunogenicity, and short duration of the expression limits their application specifically for the gene therapy of inherited diseases.

AAV vectors are currently particularly prominent in the field of gene therapy mainly because of their low immunogenicity and the maintenance of their expression for over 10 years when the cells do not divide. Therefore, several gene therapy drugs using AAV vectors have been approved for clinical use [20,21,22,23,24,25]. The maximum capacity of an AAV vector is 4.7 kb. This size is often sufficient for gene therapy but is not necessarily sufficient for genome editing therapies because Cas9 derived from *Streptococcus pyogenes* (spCas9), the most commonly used Cas9, is very large. The coding region of Cas9 is 4.2 kb, so the size of its expression unit normally exceeds the capacity of AAV. Additionally, not only the expression unit(s) of guide RNA (gRNA, hereafter) but also a donor DNA for correction of a mutated gene are required. For this reason, AAV vectors must employ a shorter Cas9 derived from *Staphylococcus aureus* (saCas9), which is 1.6 kb in length [26,27]. However, this Cas9 cannot be used for the safer double-nicking strategy, as explained below.

Off-target effects are the most serious problem in genome editing therapy. A strategy to combat this called double nicking is available, which uses Cas9 nickase (Cas9n), a variant of spCas9, introducing a nick instead of a cleavage and reduces off-target effects by up to 1500-fold [28]. In this strategy, two nicks are introduced by Cas9n using two gRNAs, and the length between the 5′ terminals of the two gRNAs present in the top and bottom strands must be between nucleotides (nt) 30 and −5 for efficient cleavage. Because one gRNA specifies 20 nt sequences, with 40 nt sequences in total, the cleavage specificity is doubled. In addition, off-target nicks produced by Cas9n are immediately repaired [29]. Therefore, the double-nicking strategy can, at least in part, solve the problem of off-target effects. Although double-nicking cleavages cause chromosomal rearrangements similarly to native Cas9, the double-nicking strategy is preferable because it reduces the formation of insertions/deletions (indels) at off-target sites, which is inevitable for native Cas9.

However, the double-nicking strategy is not currently widely used because two gRNAs are required for one cleavage. If the same promoter is used for the expression of two gRNA units, deletion of the unit due to homologous recombination is normally unavoidable in the steps of both vector construction in *Escherichia coli* and viral vector amplification in human cells. Another reason for the unpopularity of the double-nicking strategy is that AAV vectors cannot use a double-nicking strategy because they must use saCas9 owing to the size limitation. The PAM sequence, essential for Cas9 cleavage, must be present within 1 nt from the 20 nt target sequence. The PAM sequence of spCas9 consists only of dinucleotides of 5′-NGG-3′ and, because the dinucleotide “GG” is frequently present in the cell genome, a possible cleavage site of double nicking, where two GGs in the different strands are by chance present within 30 nt, is often available. Meanwhile, the PAM sequence of saCas9 used in the AAV vector is 5′-NNG(A/G)(A/G)T-3′, so possible cleavage sites are rare. Consequently, AAV vectors can hardly use the double-nicking strategy.

We have shown that multiplex eight guide RNA units are stably maintained in the AdV genome without deletion, even though the units are completely identical, only with the exception of the 20 nt target sequences [30]. Therefore, surprisingly, homologous recombination causing deletion of the units hardly occurs, although AdV genomes are replicated to 10 million copies in a single 293 cell. Thus, the limiting step in the construction of AdVs possessing eight multiple gRNA units is, unexpectedly, not the viral replication step in 293 cells, but the step of replication of large plasmids carrying the AdV genome in *E. coli*. We routinely obtain large plasmids (called cosmids) with no deletion of the eight units by using in vitro packaging of lambda phage, where deleted plasmids of smaller sizes are removed through size selection. We also developed a “tetraplex tandem” method that allows the construction of cosmids possessing AdV genomes bearing four and eight multiplex gRNA units in a single step [31,32].

AdVs simultaneously expressing four and eight gRNAs (4 g AdVs and 8 g AdVs, respectively, hereafter) are particularly desirable. Using two gRNAs, the double-nicking method can be employed and, additionally, simultaneous cleavage at two sites and subsequent end-joining cause an irreversible deletion, resulting in complete gene disruption in one step. Therefore, 4 g AdVs can safely and efficiently destroy the target gene, as shown by Nakanishi et al. [31]. However, an unavoidable problem with the strategy of 4 g AdVs is that, if the activity of one gRNA is low, this strategy using four gRNAs does not work well. Moreover, a mismatch of only 1 nt in 80 nt target sequences of four gRNAs in total abolishes this strategy, which often occurs in practical applications. In this context, 8 g AdVs, which can be constructed in one step using the tetraplex tandem method, solve this problem and are clearly more effective than 4 g AdVs. The 8 g AdVs are particularly useful for the therapy of infectious diseases because genome sequences of pathogens are generally heterogeneous. In fact, we showed that an 8 g AdV effectively disrupted the hepatitis B virus genome, where four gRNAs could not cleave the genome because of mismatches, but the remaining four gRNAs efficiently cleaved the genome through irreversible deletion [30].

In genome editing experiments, AAV vectors and AdVs generally employ a co-infection strategy to account for size limitations or technical difficulty, as explained below. In the co-infection strategy, a cleavage occurs only when both gRNA-expressing vector and Cas9-expressing vector are simultaneously introduced into a single cell. In particular, under conditions where the two vectors are diluted, such as in vivo, the number of cells infected with both vectors will exponentially decrease. In contrast, if an all-in-one vector is used, the Cas9(n) enzyme and gRNAs are always present at the same time. Therefore, it is expected that all-in-one AdV is more effective than co-infecting AdVs.

All-in-one AdV expressing eight multiplex gRNAs (8 g all-in-one AdV, hereafter) is attractive, but construction of the vector has been considered challenging. To date, very few reports of all-in-one type AdVs have been published [33,34], and the maximum number of guides expressed was two [34]. Besides E1- and E3-deleted AdVs, a helper-dependent (HD) AdV lacking all of the adenovirus genes has been reported, which is an all-in-one type and possesses both three gRNA units and a Cas9 expression unit [35,36,37]. However, the construction of an HD vector is not simple because it requires a helper AdV supplying all viral gene products in trans, and the large-scale production necessary for their therapeutic application is difficult.

Two obstacles, at least, are expected for successful construction of all-in-one AdVs containing four gRNA units (4 g all-in one AdVs, hereafter) and an 8 g all-in-one AdV. First, the production of both 4 g and 8 g all-in-one AdVs is expected to be difficult because, although all four gRNA units can be maintained under the normal amplification conditions in 293 cells in the co-infection strategy, extremely large amounts of Cas9n and gRNAs are produced from 10 million copies of the AdV genome in the all-in-one strategy. The genome of 293 cells may be damaged by off-target cleavages, which induce homologous recombination and deletion of the units. Second, the 8 g all-in-one AdVs are associated with an additional difficulty: the size of the Cas9n expression unit and eight gRNA units in total exceeds the capacity of the normal AdV backbone currently available. Therefore, we must develop a new vector with a larger capacity.

We here described obstacles for construction of 4 g and 8 g all-in-one AdVs. Surprisingly, the 4 g all-in-one AdVs could be obtained with no lack of gRNA units in spite of extreme expression of both Cas9n and gRNAs in 293 cells. To obtain larger capacity for construction of 8 g all-in-one AdV, E4/E3 deletion in the AdV backbone was enlarged and the U6 promoter was shortened. The deletion of gRNA units of 8 g all-in-one AdV was, in fact, observed but the vector was still sufficient for use. The 4 g and 8 g vectors successfully cleaved the expected two and four double-nicking target sites. However, unexpectedly, the genome editing efficiencies of both AdVs were not higher than expected considering those of co-infecting AdVs, suggesting that a third obstacle was present. Possible reasons for the low genome editing efficiency are discussed.

## 2. Results

### 2.1. Enlargement of E3/L4 Deletion in the Vector Backbone

The 4 g all-in-one AdV contains both four gRNA expression units and a Cas9n expression unit (Figure 1, top), and the 8 g all-in-one AdV contains both eight gRNA units and a Cas9n unit (middle). The size of the final 8 g all-in-one AdV was calculated to be 39.86 kb (110.9% of the wild-type adenovirus genome), if we employed the normal human U6 promoter and the original Adex backbone [38]. This size considerably exceeds the reported upper limit of the AdV genome, which is strictly 37.7 kb (105% of the wild-type adenovirus genome) [39]. In a widely used vector developed by Bett et al. [40], the 5′ end position of the L4/E3 deletion is the BglII site (Figure 2a, fourth row, AxdB in blue). When we use the BglII site for deletion, the calculated size of the 8 g all-in-one AdV is 38.02 kb (105.8%) even if we used the shortened U6 promoter described later. Because this length still exceeds the reported packaging limit of 37.7 kb, we attempted to enlarge the L4/E3 deletion further. Bett et al. [39] also described that the AdV containing an E3/L4 deletion that starts at the initiation codon ATG of the putative 12.4K-URF gene (first row, boxed ATG; URF denotes unidentified reading frame) was not available. Therefore, we enlarged the deletion up to the ApaI site (second row, AxdA), present 153 bp upstream from the BglII site of AxdB. The final size of the 8 g all-in one vector becomes 37.87 kb (105.4%). Although this size still slightly exceeds the reported packaging limit, we adopted the ApaI site because an AdV of a larger size has been reported (see Section 3).

To examine the influence of the E3 ApaI deletion, we constructed AxdA-oCBNC9 (Figure 1, bottom), which possesses the L4/E3 ApaI deletion and only contains the Cas9n expression unit at the E1 region (Figure 2b, shown as AxdA). We also constructed AxdB-oCBNC9 as a control, which is identical to AxdA-oCBNC9 but possesses the BglII deletion instead of the ApaI deletion (shown as AxdB). The titers of AxdA-oCBNC9 and AxdB-oCBNC9 were 3.2 × 10^9^ (±3.5 s.d.) relative viral titer (rVT, equivalent to TCID_50_) and 6.4 × 10^9^ (±1.0 s.d.) rVT, respectively. Although this titer of AxdA-oCBNC9 is approximately half that of AxdB-oCBNC9, it is sufficient for experiments in vitro and in vivo.

To examine the activity of Cas9n expressed from these vectors, Gx11 cells [41], which are derived from HepG2 cells containing the X region of hepatitis B virus (HBV) integrated into the cell genome as a source of target sequences, were infected with these AdVs together with Axg8HBV-X-KK expressing four pairs of guide RNAs that cleave four sites of the HBV-X gene by double nicking (Appendix A). Unexpectedly, AxdA-oCBNC9 showed very low efficiency of double-nicking disruption (Figure 2c): the amount of intact HBV-X DNA using AxdB decreased 26.2% at a multiplicity of infection (MOI) of 10, while that using AxdA decreased 5%, only one fifth of AxdB (MOI of 10, yellow bars). The only difference is that the former loses the C-terminal half of the coding region of the putative 12.4K-URF gene, while the latter possesses most of this gene (Figure 2a, region from green “12.4K gene start” in the first row, amino acid sequences and “12.4K gene end” in the last row). To date, no reports have been published about the function of the putative 12.4K-URF gene, so this is the first report of its kind. However, because our purpose is to test whether 8 g all-in-one AdV with no deletion of these gRNA units can be constructed, and because this is the only backbone possessing a sufficient capacity to allow us to construct the 8 g all-in-one AdV, we used this vector backbone containing the L4/E3 ApaI deletion. Note that 4 g all-in-one AdVs possess this enlarged L4/E3 deletion commonly to 8 g all-in-one AdV (Figure 1, top and middle).

### 2.2. Construction of All-in-One AdVs Expressing Four gRNAs

We attempted to construct 4 g all-in-one AdVs (Figure 1, top). These AdVs possess the L4/E3 ApaI deletion and their genomes are sufficiently short for efficient packaging. Both Axda4g-mH2Aa-oCBNC9 and Axda4g-mH2Aa-iCBNC9 (outward 4 g AdV and inward 4 g AdV, respectively, hereafter) contain the Cas9n expression unit under the CB promoter at the E1 insertion site and an array of four gRNA expression units, with the original U6 promoter, targeting the mouse *H2-Aa* gene [31] at the E4 insertion site. These two AdVs are identical except for the orientations of the Cas9n expression unit: The former, outward 4 g AdV, and the latter, inward 4 g AdV, contain the Cas9n unit in outward and inward orientations, respectively. The 293 cells were transfected with the linearized genomes of these AdVs. The growth of the viral clones was unusually slow and it took 3 weeks to reach full lysis of 293 cells owing to viral growth, while approximately 2 weeks is normally sufficient. We prepared total cellular DNA of second stocks and performed digestion with BspEI to examine both the structure of the viral genome and the deletion of the gRNA units. Because the copy number of viral genomes is very high, viral DNA fragments can be seen directly without the purification of viral DNA (Figure 3). The intensities of the bands were weaker than those of normal AdVs, suggesting that the copy number of the viral genomes significantly decreased. Remarkably, however, the intensity of the 2.2 kb band containing an intact array of four gRNA units hardly decreased compared with that of the 2.4 kb band derived from the AdV backbone (inward, blue arrow and “A” above). The result was also confirmed for the outward AdV. These results suggest that four multiplex gRNA units were almost completely stable on the all-in-one AdV genome, although viral growth was significantly delayed.

We prepared samples from the third stock to the purified sixth stock for use in the in vivo experiment, and found that the four gRNA units were maintained with almost no deletion even in the purified stock of both outward and inward AdVs (upper and lower panels, right). The titers of the fifth stocks of outward and inward AdVs were almost the same, 1.7 × 10^9^ (±1.0 s.d.) rVT and 1.6 × 10^9^ (±1.0 s.d.) rTV, respectively, which were approximately half of those of the AxdA-oCBNC9 lacking four gRNA units. These results may suggest that the expected off-target cleavages caused slow growth and a decreased number of viral genomes but, unexpectedly, did not cause loss of the gRNA units.

### 2.3. Preparation of All-in-One AdV Containing Eight gRNA Expression Units

Because the length of 8 g all-in-one AdV genome exceeds the packaging limit, we constructed a shortened U6 promoter (Figure 4). This promoter consists of three parts. The distal and proximal sequence elements (DSE and PSE, respectively; two red boxes in the first and third rows), essential for promoter activity, are from −241 to −214 and from −66 to −47, in the nucleotide positions, respectively. The region between DSE and PSE negatively regulates transcription and, when this region is deleted, the promoter activity is maintained or even increases [42]. Therefore, we constructed a shortened U6 promoter lacking a sequence of 126 nt between nt positions −219 and −84 (boxed in blue). Consequently, the U6 promoter lacks the region from 265 bp to 139 bp (approximately 52% of its total length), and the size of one unit decreases to approximately two-thirds, from 368 bp to 268 bp.

To obtain 8 g all-in-one AdV, Axda8sg-mNTCP-oCBNC9 (Figure 1, middle), the AdV genome in the cosmid was excised and transfected to 293 cells to obtain AdVs. Similarly to the case for the 4 g vector, it took approximately 3 weeks for full lysis to be reached. The second stocks of the viral clones were prepared, and the total cellular DNAs of the infected 293 cells were digested with BspEI (Figure 5, left). The results were markedly different from those of 4 g AdVs: the 3.0 kb bands derived from the intact array of the eight gRNA units with the shortened U6 promoter were faint or disappeared in four out of six clones (blue arrow at 3.0, lanes 1, 2, 3, and 5). Alternatively, bands corresponding to arrays containing fewer copies of gRNA units were observed (shown as asterisks). Therefore, deletions through homologous recombination frequently occurred in the case of 8 g AdVs. For example, in clone 2, the intact 3.0 kb was not seen but, in turn, an extra 1.8 kb band corresponding to the array containing four gRNA units was observed (shown by a hash symbol). Because the intensity of the extra band was similar to that of the 1.7 kb band of the vector backbone, the result may suggest that a deletion involving the loss of four gRNAs occurred at a very early stage, and no further deletion occurred until the end of full lysis, as observed for 4 g all-in-one AdV.

Nevertheless, one clone, clone 6, produced a strong 3.0 kb band of eight intact units (lane 6). Remarkably, considering that the 3.0 kb band was much more intense than that of 2.6 kb (band “A” below the 3.0 kb band), which is 87% of its size, the array of eight units appeared to be maintained almost intact without deletion. The viral stock of this clone was amplified, and the total cellular DNAs from the third stock to the purified stock were digested with BspEI. Contrary to the findings for the second stock, the ratio of the intensities between the 3.0 kb band of eight units and the 2.6 kb band of the vector backbone was reversed (left panel, lane 6, and right panel, lanes “4th” to “purify”). In the lane of purified stock (lane “purify”, bold blue arrow), the intensity of the bands of 3.0 was approximately one-third or slightly more compared with that of the band “A” of 2.6 kb, suggesting that the molecular ratio of the array of the eight units to the vector backbone was approximately one-third. The purified stock also contained deleted arrays of four and three gRNA units (thin arrows). Therefore, although all-in-one AdV maintaining eight gRNA units was actually obtained, the eight units were much less stable than the four units in all-in-one AdVs. Nevertheless, the 8 g all-in-one AdV stock could practically be used for experiments because, even when the ratio of the eight intact gRNA units to the vector backbone was one-third, sufficient experimental results could be obtained upon simply using three times more AdVs. In addition, because a vector containing a deleted array of units numbering less than eight, which occupied approximately two-thirds of the total, was not counted but expressed gRNAs from the remaining gRNA units, this vector stock must be more effective than the stock only consisting of the vector possessing the intact array and not containing deleted arrays.

The 8 g all-in-one AdV shown above possesses the Cas9n expression unit in the outward orientation. When six clones in the second stocks of inward AdV were tested, no AdV maintaining eight intact units was present (Appendix A), suggesting that the examination of many clones may be necessary to obtain 8 g AdV stock containing eight intact units.

### 2.4. Genome Editing of Target Gene Using 4 G All-in-One AdVs

The sequences and positions of the four gRNAs used for examination of the genome editing efficiency of 4 g all-in-one AdVs targeting the mouse *H2-Aa* gene are shown in Figure 6a. MEF cells were infected with 4 g outward and inward AdVs (Figure 1, top) at MOIs of up to 300, and the target region of 0.56 kb in length was amplified by PCR (Figure 6b, outward and inward, 0.56 kb bands). Because double-nicking cleavages occurred at gRNA 1/gRNA 2 (Figure 6a, second row) and at gRNA 3/gRNA 4 (third row), ~0.46 kb DNA possessing a deletion between gRNA 1/gRNA 2 and gRNA 3/gRNA 4 was produced by non-homologous end-joining (Figure 6c). Detection of the 0.46 kb band showed that all four gRNAs were active. The intensity of the intact 0.56 kb band decreased depending on the MOIs (Figure 6b), and the deletion efficiencies at MOI of 300 were 55% and 47% for outward and inward, respectively. Note that the genome editing efficiency was higher than the deletion efficiency because double-nicking cleavages generate not only large deletions but also small indels, which are undetectable in the PCR deletion assay.

To examine the efficiency of target genome editing in vivo, a total of eight newborn mice were intravenously administered with AdVs at 2.5 × 10^8^ rVT. Total liver DNA was extracted and the indels were detected using the T7E1 assay, where the 0.56 kb fragment was cleaved at the position of indels (Figure 6c). Both on day 7 and on day 15, bands of 0.33 kb and 0.16 kb produced by cleavage at the position of double nicking were observed below the intact 0.56 kb band (Figure 6d, left and right panels). The broad band of 0.16 kb indicates the DNA fragment from the 5′ end of the forward primer to the heterologous cleavage site of gRNA 1/gRNA 2 (Figure 6a, second row). In addition, the broad 0.33 kb band indicates the fragment from gRNA 3/gRNA 4 to the 5′ end of the R primer. The indel rates were 14% to 16% on day 7 and 19% to 21% on day 15 (Figure 6d, left and right panels, lanes +).

When adult mice were administered with outward and inward AdV, the deletion efficiency (Figure 6e, left) and indel rate (right) on day 14 were up to 26% and 32%, respectively. The indel rates in the T7E1 assay were higher in adult mice than in neonatal mice (26% to 32% vs. 19% to 21%). This finding can be explained by most hepatocytes in the liver of adult mice being quiescent and not dividing, so a high copy number of the AdV genomes can be stably maintained.

### 2.5. Genome Editing of Targeted Genes Using 8 G All-in-One AdV

The sequences and positions of eight gRNAs targeting the mouse *NTCP* gene are shown in Figure 7a. MEF cells were infected with the fifth stock of 8 g all-in-one outward AdV at the indicated MOIs. Total cellular DNA was prepared 3 days later, and deletion efficiencies were examined by amplifying the 0.65-kb target region by PCR (Figure 7b). Note that the amount of the intact array of the eight gRNA units in the fifth stock of 8 g all-in-one AdV was only slightly more than one-third of the total based on the intensity of the bands of the backbone (Figure 5, right, the 3.0 kb band in lane 5). The deletion efficiency using this stock at MOI of 300 was 39% (Figure 7b, left, lane 300). This efficiency is lower than that at MOI of 300 for 4 g all-in-one AdV (i.e., 47%), but higher than that at MOI of 100 (i.e., 32%) (Figure 6b, lanes 100 and 300). Therefore, although the amount of intact 8 g AdV was less than one-third of the total in the stock, the AdV stock in the current form is practically more effective if three times more stock is used.

In addition to the 0.65 kb band, a broad band of approximately 0.5 kb and a region of approximately 0.3 kb were also observed (Figure 7b, right panel, longer exposure of the left panel). As shown in the schema in Figure 7b (lower right), the 0.5 kb bands observed at lanes 3 to 300 were generated by double cleavage of gRNA 3/gRNA 4 (g3/g4, hereafter) and g7/g8. In addition, the 0.3 kb band (lanes 100 and 300) can be produced by the double cleavage of g1/g2 and g7/g8. Meanwhile, the 0.60 kb band expected to be produced by the double cleavage of g3/g4 and g5/g6 or the 0.55 kb band produced by the double cleavage of g5/g6 and g7/g8 was not observed. Because g5/g6 is common, the cleavage by this pair appeared inefficient for production of the bands of 0.6 kb and 0.55 kb. The low efficiency of g5/g6 double nicking could be explained by the partial overlap between gRNA 4 and gRNA 5 (Figure 7a, fourth row, right end, blue vertical arrow), which can reduce the binding efficiency of Cas9n to gRNA 5. Therefore, with the exceptions of gRNA 5 and gRNA 6, all other gRNAs worked in the 8 g all-in-one AdV. T7E1 assay was also performed, and the indel rate at MOI of 300 was 42% (Figure 7c, lane 300+). The size and intensity using the T7E1 assay agreed with those using the PCR deletion assay.

To examine the efficiency of genome editing in vivo using 8 g all-in-one AdV, 0.8 × 10^9^ rVT of the AdV was administered intravenously to newborn mice, and total hepatocyte DNA was extracted on day 15. The PCR assay yielded broad bands of 0.5 kb and 0.3 kb similar to the in vitro results shown in Figure 7b, with deletion efficiencies of 9% and 5% (Figure 7d, left, lanes -m1 and -m2). In addition, the indel rates of T7E1 assay were 25% and 18% (right, lanes -m1 and -m2). Therefore, with the exceptions of gRNA 5 and gRNA 6, all other gRNAs functioned in 8 g all-in-one AdV in vivo as well as in vitro.

## 3. Discussion

We here showed that all-in-one AdVs simultaneously expressing four gRNAs can be successfully constructed. Because the AdV genome was amplified at 10^9^ copies per cell in 293 cells and vast amounts of both gRNA and Cas9n were produced, we expected that the cell genome was probably damaged by off-target cleavages, leading to the activation of repair systems and enhancement of homologous recombination. We had thought that four gRNA units in the all-in-one AdV would possibly be deleted and successful construction would probably be difficult. Although we used a double-nicking strategy, which was reported to reduce off-target effects caused by native Cas9 by up to 1/1500-fold in transfection experiments [28], the reduction in off-target effects that we observed was actually only three orders of magnitude and appeared insufficient to completely prevent activation of the repair systems. However, contrary to our expectations, the results obtained using 4 g all-in-one AdV showed that, although the full lysis in the amplification of AdV was delayed from 2 to 3 weeks, no significant deletion of multiple gRNA units was observed. This suggests that, unexpectedly, the induction of homologous recombination by off-target cleavage of the 293 cell genome did not occur.

It is currently unclear why homologous recombination was not induced in this setting. However, a simple and plausible possibility is that, even when very large amounts of the enzyme and gRNAs were present at high concentrations, Cas9n caused much less off-target cleavage than we expected. If so, nicks produced by Cas9n were very immediately repaired, so Cas9n may be much safer than we imagined. Another possible mechanism for the deletion is that, when the AdV genome replicates, the progressing 5′ end dissociates and hybridizes to the next or a more distant gRNA unit, resulting in deletions between them. However, no deletions were actually observed, indicating that this process did not occur, at least in the 4 g all-in-one AdV. Because four multiplex gRNA units in 4 g all-in-one AdV were unexpectedly stable, 4 g vectors can be used for applications where four gRNAs are sufficient.

We also showed that an 8 g all-in-one AdV, Ax8sg-mNTCP-oCBNC9, can be obtained, although deletions were observed in contrast to the case for 4 g all-in-one AdV. However, we previously reported that, in cases of co-infection when Cas9n was absent, eight multiplex gRNA units were stably maintained in the AdV genome [30]. Two different reasons for this can be considered. One obvious reason is that, in the presence of Cas9n, one of the two conditions mentioned above could have occurred because eight units is too many to be stably maintained. If this is the reason, it may be difficult to construct an 8 g all-in-one AdV possessing eight multiplex units. The other reason is that the genome size of this particular 8 g all-in-one AdV was approximately the same or slightly larger than the reported packaging limit of the AdV genome. If this is the reason for the instability, this problem should be resolved when the size of the AdV genome is further reduced. The total size of the 8 g all-in-one AdV presented here is 37.87 kb in length (105.4% of the wild-type adenovirus genome). Bett et al. [39] reported that the upper limit is 37.7 kb (105% of the AdV genome) based on the result that an AdV, AdLacZ, of 37.87 kb (105.4%) showed deletion; this size is the same as that of the 8 g all-in-one AdV described here. However, Ghosh-Choudhury reported that Ad5in52 of 38.0 kb, corresponding to 105.7% of the AdV genome, was stably produced [43]. Therefore, it is currently unclear which explanation for the instability observed in the case of the 8 g all-in-one AdV is valid.

Upon using 4 g all-in-one AdVs of Axda4g-mH2Aa-oCBNC9 and Axda4g-mH2Aa-iCBNC9 at MOI 100, the efficiency of destruction targeting the mouse *H2-Aa* gene in MEF cells was 38% and 32%, respectively (Figure 6b, outward and inward, lanes 100). However, the observed efficiency was unexpectedly low compared with that upon co-infection. Ax4g-mH2Aa is the parent AdV containing the same four gRNA units, but lacks a Cas9n expression unit, and 81% efficiency was reported upon co-infection with Cas9n-expressing AdV, both at an MOI of 100 [31]. Therefore, the destruction efficiency of 4 g all-in-one AdV was much lower than that of co-infection with Ax4g-mH2Aa and AxCBNC9. The structural difference between the two methods is that 4 g all-in-one vectors used the AdV backbone lacking the E3 region: the 5′ position of the E3 deletion of g4 all-in-one AdV starts at position 27,980 (Figure 2a, second row, AxdA), while that of Ax4g-mH2Aa starts at position 28,593 (outside of Figure 2a), covering the whole of the 12.4K gene. Because deletion of 153 bp between AxdA and AxdB caused low genome editing efficiency (Figure 2c), the result suggested that the lower genome editing efficiency of the all-in-one AdV than that of co-infecting AdVs was probably caused by this deletion. Therefore, we could not show expression efficiency of all-in-one AdV that was higher than that of co-infection, probably because of low efficiency caused by deletion of the N-terminal half of the 12.4K gene. The mechanism behind the low efficiency is currently unclear because the function of the 12.4K gene has not been reported. It is desirable to develop a vector backbone that avoids the low efficiency and increases the capacity to construct 8 g all-in-one AdVs. (We later succeeded in solving the problem of this low efficiency while further enlarging the E3 deletion, but the results will be published elsewhere). Mizuguchi et al. reported a higher-capacity AdV possessing an L4/E3 deletion 292 bp larger than that of the AxdA [44]. This vector solves the size limitation problem to construct 8 g AdV, but our results may suggest that deletion of the whole of the 12.4K gene could cause low efficiency of genome editing similarly to our vectors using the AxdA backbone, if an all-in-one AdV is constructed using their L4/E3 backbone.

In conclusion, we successfully constructed a 4 g all-in-one AdV simultaneously containing four multiplex gRNA units and a Cas9n expression unit, and demonstrated that this vector can be amplified to levels sufficient for in vivo experiments with no deletion of the units. We also constructed an 8 g all-in-one AdV, although the amount of intact AdV retaining all eight gRNA units was approximately one-third in the purified stock. Both g4 and g8 all-in-one AdVs showed all expected cleavages in vitro and in vivo to disrupt target genes by double nicking. However, we could not show higher genome editing efficiency of all-in-one AdV than that of co-infecting AdVs in this study. The results described here provide a basis for the development of effective and safe all-in-one AdVs that can be valuable for genome editing therapy.

## 4. Materials and Methods

### 4.1. Cell Culture

The human 293 (ATCC CRL-1573) and HepG2 (ATCC HB-8065) cell lines were derived from human embryonic kidney and human hepatocellular carcinoma, respectively. Gx11 is a cell line derived from HepG2, which possesses the HBV X gene in an inactive state [41]. Mouse embryonic fibroblast (MEF) cells were derived from C57BL/6 mice for use in this study [45]. The 293 cells were cultured in Dulbecco’s Modified Eagle’s Medium (DMEM) (Kohjin Bio, Co., Ltd., Saitama, Japan) supplemented with 10% fetal calf serum (FCS). The 293 cells constitutively express adenoviral E1 genes and support the replication of E1-substituted AdVs. The HepG2 and Gx11 cells were kept in high-glucose DMEM (Kohjin Bio) supplemented with 10% FCS.

### 4.2. Construction of Cosmids Containing AdV Genome Bearing Multiplex gRNA Units

Multiplex eight gRNA expression units targeting the mouse *NTCP* gene were constructed in accordance with the Tetraplex-guide Tandem method [32] using the oligonucleotides shown in Appendix A. The sequence of the head-block reverse primer, ampHeadTE-BxSs-R, is 5′-gcg AATATT CCACTCTTCTGG (BstXI) AGCT GTTGACGCCAGCAACTGTACA GGATCC-3′ (underline shows sequence homologous to pParent head-E), and the sequence of the mid-block forward primer, ampMidTA-BxSs-F, is 5′-gcgAATATTG CCAGAAGAGTGG (BstXI) AGCT GTCAAC GGCGTCAGTTGCTG GCTAGC-3′ (underline shows sequence homologous to pParent mid-A), which were used instead of ampl Head-A R and ampl Mid-A F. For ligation of head block and mid-block, BstXI was used instead of AlwNI to obtain more efficient ligation. The PCR products of head block and mid-block were doubly digested with BstXI and PvuII and cloned into the SwaI site of pAxa4w-o/iCBNC9. Multiplex four gRNA expression units targeting the mouse *H2-Aa* gene [31] were amplified by PCR and also cloned into the SwaI site of pAxa4w-o/iCBNC9.

### 4.3. Production of AdVs

AdVs were produced as previously described [31,32]. Briefly, 293 cells were transfected with the BstBI-linearized AdV genome in pAxc4wit2, using Lipofectamine LTX with Plus Reagent (Thermo Fisher Scientific, Inc., Waltham, MA, USA). The next day, the cells were transferred to a 96-well plate. The first viral stocks were obtained within 2 weeks (approximately 150 µL). Cells in the 24-well plates were infected with half of the first stock to obtain the second stock, and one-tenth of the second stock was used for the third stock. AdVs used for mouse experiments were purified using a two-step CsCl gradient in accordance with the method of Kanegae et al. [46]. AdVs were titrated using HepG2 by a method described previously. The copy numbers of the viral genome were measured by real-time PCR. The relative viral titer (rVT) in this reference is normally equivalent to TCID_50_ (50% tissue culture infectious dose) and PFU (plaque-forming unit).

### 4.4. Conventional PCR

A single cleavage produces very small indels, which cannot be detected by conventional PCR, while two simultaneous cleavages by double nicking produce a large deletion between them (bridged deletion), which can easily be detected using conventional PCR. MEF cells in 60 mm-well cell culture dishes were infected at the indicated MOIs and incubated for 3 days postinfection in DMEM supplemented with 5% FCS. Total cellular DNAs were prepared and amplified by PCR with Tks Gflex DNA polymerase (Takara Bio Inc., Kusatsu, Japan) using the primers indicated in Figure 6a and Figure 7a. The PCR cycling conditions were as follows: 94 °C for 1 min, followed by 30 cycles at 98 °C for 10 s, 65 °C for 15 s, and 68 °C for 30 s. Images of DNA agarose gels after electrophoresis were recorded using a Printgraph AE-6932GXES (ATTO Corp., Tokyo, Japan) and were analyzed using the software ImageJ 1.52a version [31]. The PCR products of the genome-edited DNA smaller than the original DNA were quantified, and the deletion efficiency was calculated as ratios against the total DNA in the same lane.

### 4.5. T7 Endonuclease Assay

The T7EI assay was performed as described previously [31]. Briefly, an amplified PCR product was reannealed to form a heteroduplex using the following program: denaturation at 95 °C for 5 min, reannealing from 95 °C to 85 °C at −2 °C/s, holding at 85 °C for 1 min, cooling from 85 °C to 25 °C at −0.1 °C/s, and holding at 25 °C for 1 min, followed by cooling to 4 °C. The sample was then exposed to T7 endonuclease I (New England BioLabs Japan Inc., Tokyo, Japan) at 37 °C for 15 min and analyzed on agarose gels. These gels were imaged and quantified by ImageJ.

### 4.6. Cellular DNA Isolation

For the preparation of total cellular DNA, cells were suspended and incubated in a lysis buffer containing 10 mM Tris-HCl (pH 8.0), 150 mM NaCl, 10 mM EDTA, 100 µg/mL proteinase K, 80 µg/mL RNase A, and 0.1% SDS at 50 °C for 2 h. The mixture was extracted twice with phenol/chloroform and twice with chloroform, precipitated with two volumes of ethanol at −20 °C for 1 h, and then washed once with 70% ethanol. The pellet was dissolved with TE buffer.

### 4.7. In Vitro and In Vivo AdV Infection

Animal experiments were approved by the Institutional Committee for Animal Experiments in Institute of Microbial Chemistry (Tokyo, Japan) and performed in accordance with the relevant guidelines and regulations to minimize animal suffering. MEF cells in 6 cm cell culture dishes were infected with AdVs at the indicated MOIs and incubated for 3 days postinfection. For the in vivo administration of AdVs, 50 µL of the vectors were injected intravenously into the facial vein of newborn C57BL/6 mice (CLEA Japan, Inc., Tokyo, Japan) using a 29 G needle. The mice were then euthanized on the indicated days after administration. In the case of adult C57BL/6 mice, 100 µL of the vectors were injected into the tail vein. Total cellular DNAs were prepared and amplified by PCR with Tks Gflex DNA polymerase (Takara Bio).

## Figures and Tables

**Figure 1 ijms-25-08783-f001:**
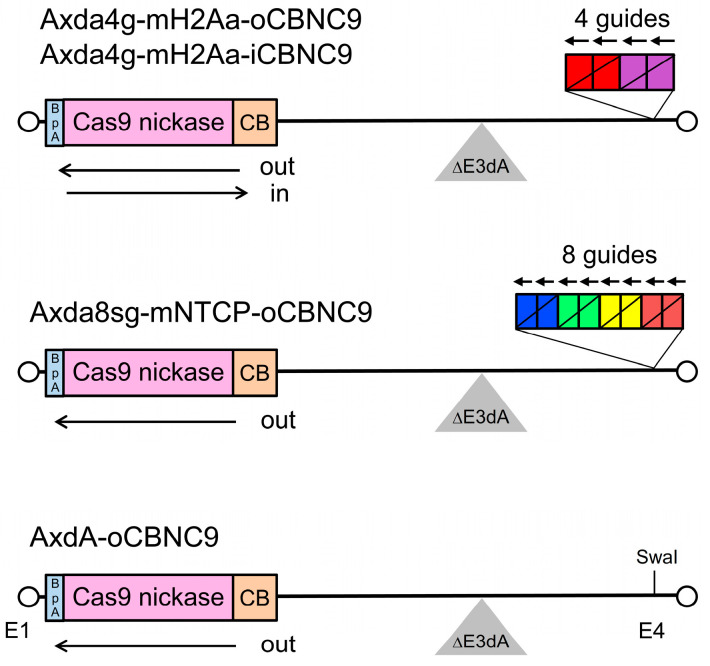
Structure of the adenovirus vectors (AdVs) used in this study. (**Top**) The 4 g all-in-one AdV. Axda4g-mH2Aa-oCBNC9 and Axda4g-mH2Aa-iCBNC9 contain a Cas9n expression unit in outward/inward orientation in the E1 region and four gRNA expression units (each box corresponds to one gRNA expression unit with the original U6 promoter and boxes in the same color indicate the double-nicking gRNA pairs) targeting the mouse *H2-Aa* gene in inward orientation within the E4 region. CB, CB promoter; BpA, bovine growth hormone polyadenylation sequence. (**Middle**) The 8 g all-in-one AdV. Axda8sg-mNTCP-oCBNC9 contains a Cas9n expression unit and eight gRNA expression units targeting the mouse *NTCP* gene in E1 and E4 regions with a shortened U6 promoter, respectively. (**Bottom**) AdV expressing Cas9n. AxdA-oCBNC9 is the parent AdV containing only the Cas9n unit.

**Figure 2 ijms-25-08783-f002:**
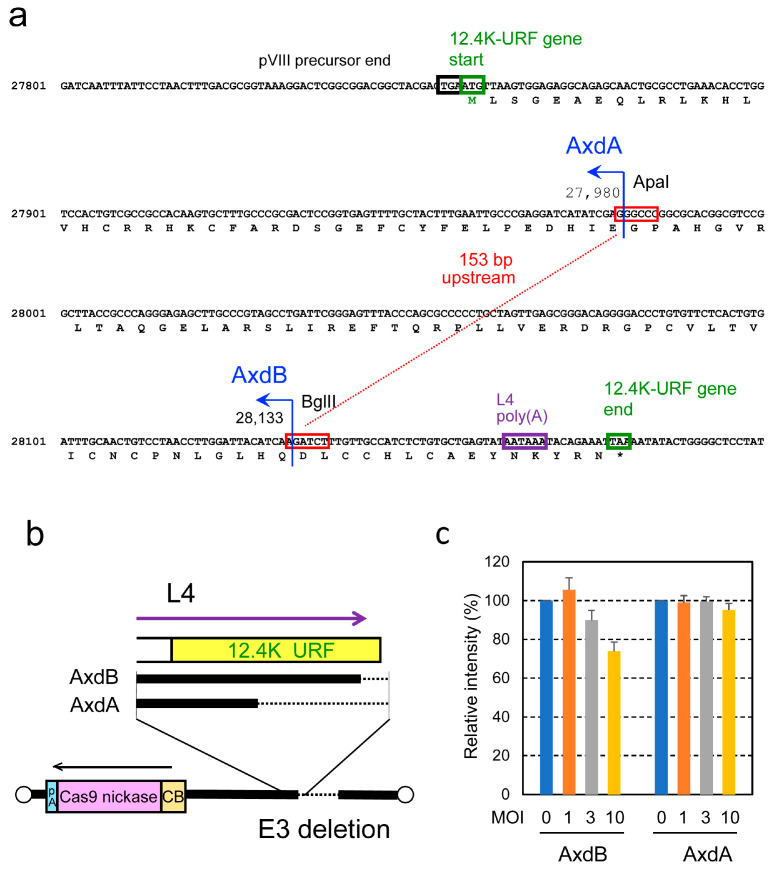
Enlargement of L4/E3 deletion in the adenovirus vector. (**a**) Sequences of the L4/E3 junction region including 12.4K-URF gene. AxdA vector possesses an L4/E3 deletion 153 bp larger than that of the widely used AxdB vector. The deletion of AxdB starts from the BglII site, while the deletion of AxdA starts from the ApaI site, located 153 bp upstream of the BglII site. Green boxes indicate the start and stop codons of the 12.4K-URF gene. Black box is the end of the pXIII precursor and purple box is the L4 polyadenylation signal. Red boxes are recognition sequences of ApaI and BglII. (**b**) Schematic diagram of AxdA-oCBNC9 and AdxB-oCBNC9, which express Cas9n under the control of the CB promoter. (**c**) Activity of Cas9n expressed from AxdA-oCBNC9 and AdxB-oCBNC9. To analyze the influence of the enlargement of the L4/E3 deletion, Gx11 cells were infected with the AdVs together with pAxc4g8 HBV-X-KK (Appendix A) and, 3 days later, total cellular DNA was extracted and PCR was performed using the primer set shown in Appendix A. The intensities of the original 0.66 kb DNA were measured using ImageJ (Appendix A) and relative intensities against MOI of 0 are shown.

**Figure 3 ijms-25-08783-f003:**
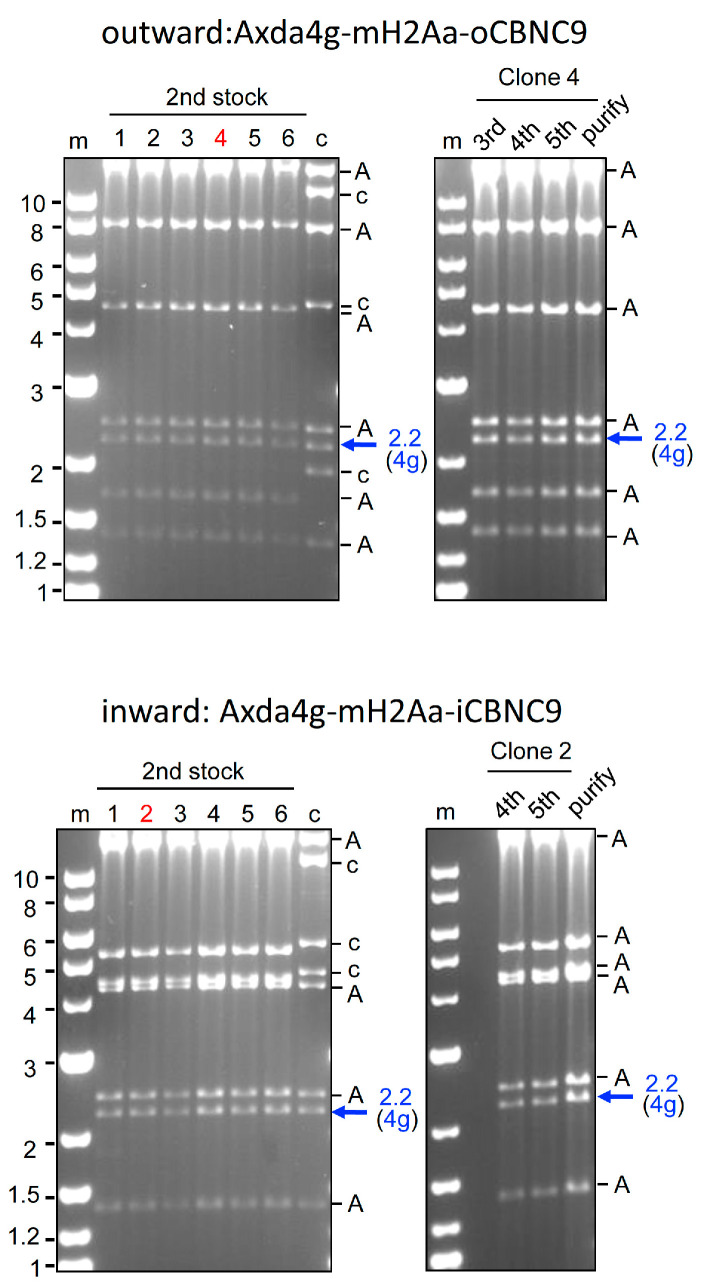
Production of all-in-one AdVs expressing four gRNAs with the original U6 promoter. The structures of Axda4g-mH2Aa-o/iCBNC9 AdV are shown at the top of Figure 1. The BspEI-cleaved AdV genomes of the second stock of six clones (**left panels**) and third (outward only), fourth, fifth, and purified stocks (**left panels**) are shown. Lane m, markers; lanes 1 to 6, clone numbers; lane c, the parent cosmid containing the AdV genome. A, AdV fragment derived from the vector backbone. A blue arrow shows the AdV DNA fragment containing four multiplex gRNA units.

**Figure 4 ijms-25-08783-f004:**
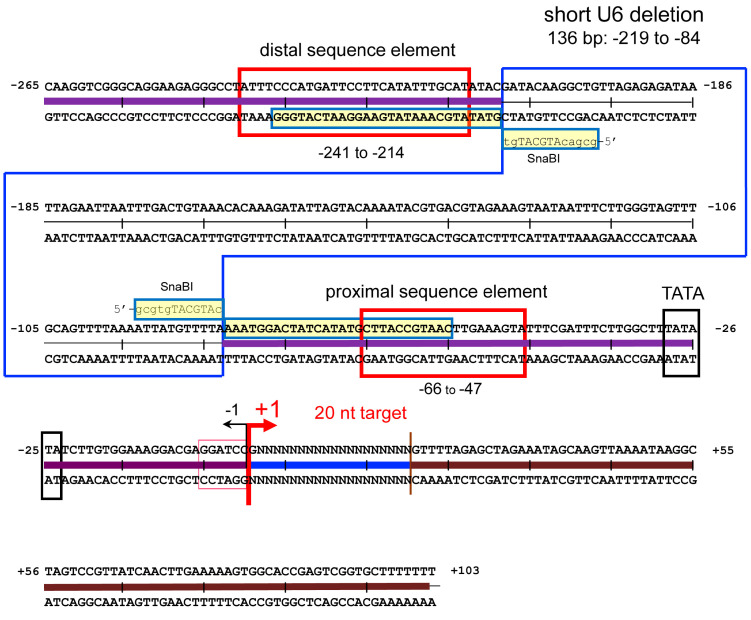
Structure of shortened U6 promoter. The DNA sequence from nt −265 to −1 shows the human U6 promoter. The shortened U6 promoter was constructed by connecting the two fragments (purple lines), one from −265 to −220 and the other from −84 to −1, with SnaBI sites. The DNA sequences from +1 to +20 indicate the target region of gRNA (blue lines) and the sequences from +21 to +103 are the gRNA scaffold (brown lines). The red boxes indicate the distal and proximal sequence elements, and the blue box indicates the deleted sequences of the original U6 promoter for construction of the short U6 promoter.

**Figure 5 ijms-25-08783-f005:**
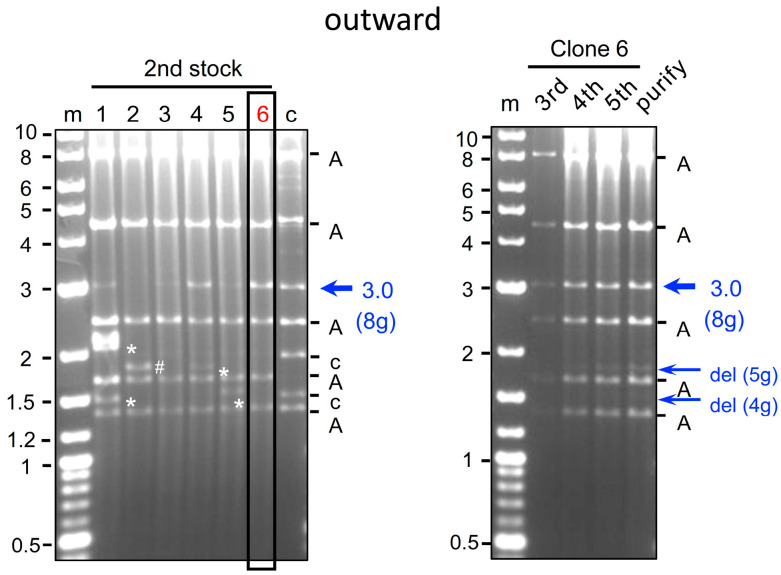
Production of all-in-one AdV containing eight gRNA expression units with the shortened U6 promoter. The structure of Axda8sg-mNTCP-oCBNC9 is shown in the middle of Figure 1. The BspEI-cleaved AdV genomes of the second stock of six clones (**left panel**) and third, fourth, fifth, and purified stocks (**right panel**) are shown. Bold blue arrows show the AdV DNA fragment containing the eight multiplex gRNA units. Thin arrows indicate bands of the DNA fragments containing four and five gRNA units. Asterisks in the left panel show bands of the DNA fragments containing fewer gRNA units. Lane m, markers; lanes 1 to 6, clone numbers; lane c, the parent cosmid containing the AdV genome. A, AdV fragment derived from the vector backbone.

**Figure 6 ijms-25-08783-f006:**
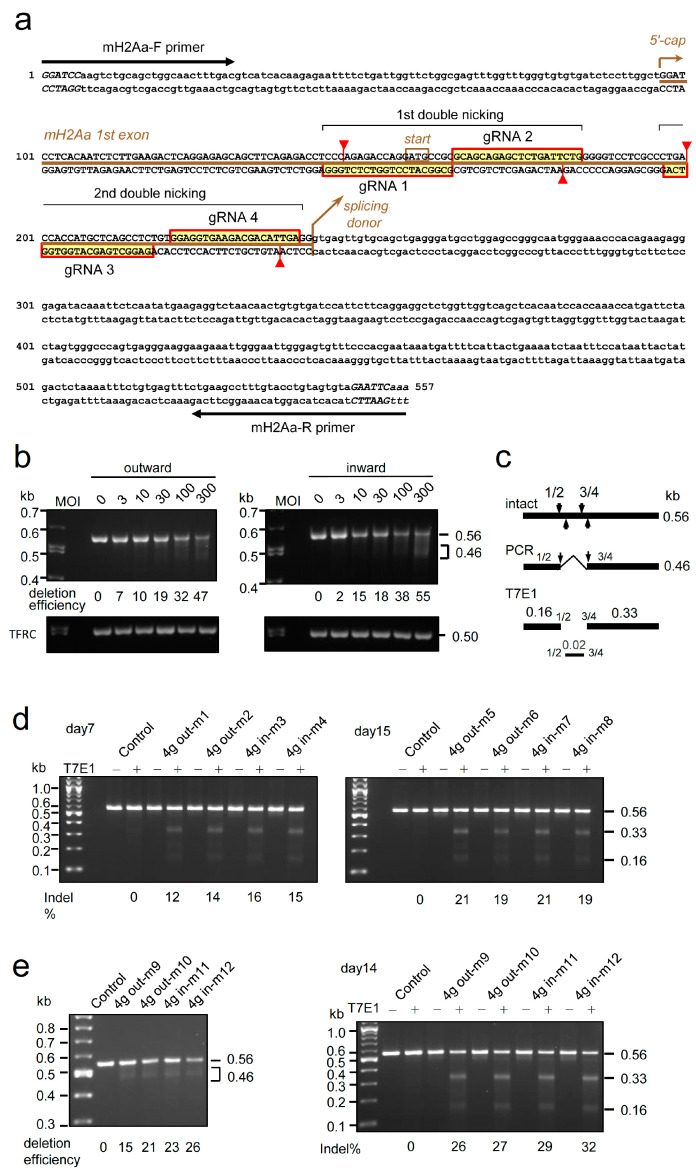
Genome editing of the mouse *H2-Aa* gene using all-in-one AdVs expressing four gRNAs. (**a**) Targeted sequences of four gRNAs expressed from Axda4g-mH2Aa-o/iCBNC9. Sequences of the first exon of the *H2-Aa* gene are shown. Bold brown line, first exon. The 20 nt recognition sequences of gRNA 1 to gRNA 4 are boxed in red. Red arrowheads show the positions of nicking by Cas9n. Long black arrows indicate the PCR primers used in (**b**–**d**). (**b**) Genome editing of the *H2-Aa* gene in vitro. MEF cells were infected with Axda4g-mH2Aa-o/iCBNC9 at MOIs of 3, 10, 30, 100, and 300. Total cellular DNA was extracted 3 days postinfection and subjected to PCR amplification using primers shown in (**a**). The PCR products of the genome-edited DNA smaller than the original 0.56 kb DNA were quantified, and the deletion efficiency was calculated as ratios against the total DNA in the same lane. The deletion efficiency, which detects only deletions but not indels, is shown under the lane numbers. As a control, the mouse TFRC gene was also PCR-amplified using the primers mTFRC-F and R (Appendix A). (**c**) Schematic explanation of the DNA fragments detected by PCR and T7E1 assay. The major PCR band of 0.46 kb in (**b**,**e** (**left**)) can be produced by deletion between guide 1/guide 2 (1/2, hereafter) and 3/4. The major T7E1 bands of 0.33 and 0.16 kb in (**d**,**e** (**right**)) can also be produced by cleavage of 1/2 and 3/4. (**d**) In vivo genome editing of neonate mouse administered Axda4g-mH2Aa-o/iCBNC9. AdVs were administered with a 2.5 × 10^8^ rVT dose into neonatal mice via the facial vein. (Three mice administered an 8 × 10^8^ rVT dose died the next day). Liver genomic DNAs were prepared 7 and 15 days after administration, and the PCR products (Appendix A) amplified by primers shown in (**a**) were subjected to T7EI assays. The control consisted of DNA from untreated C57BL/5J mice. Indel rates are shown below the lanes. (**e**) In vivo genome editing of adult mouse. Axda4g-mH2Aa-iCBNC9 and Axda4g-mH2Aa-oCBNC9 were administered via the tail vein with 1.7 × 10^10^ and 1.4 × 10^10^ rVT doses. The liver genomic DNAs were isolated 14 days after administration. The deletion efficiencies and indel rates were analyzed as above, and are shown below the lanes. The control consisted of DNA from untreated C57BL/5J mice.

**Figure 7 ijms-25-08783-f007:**
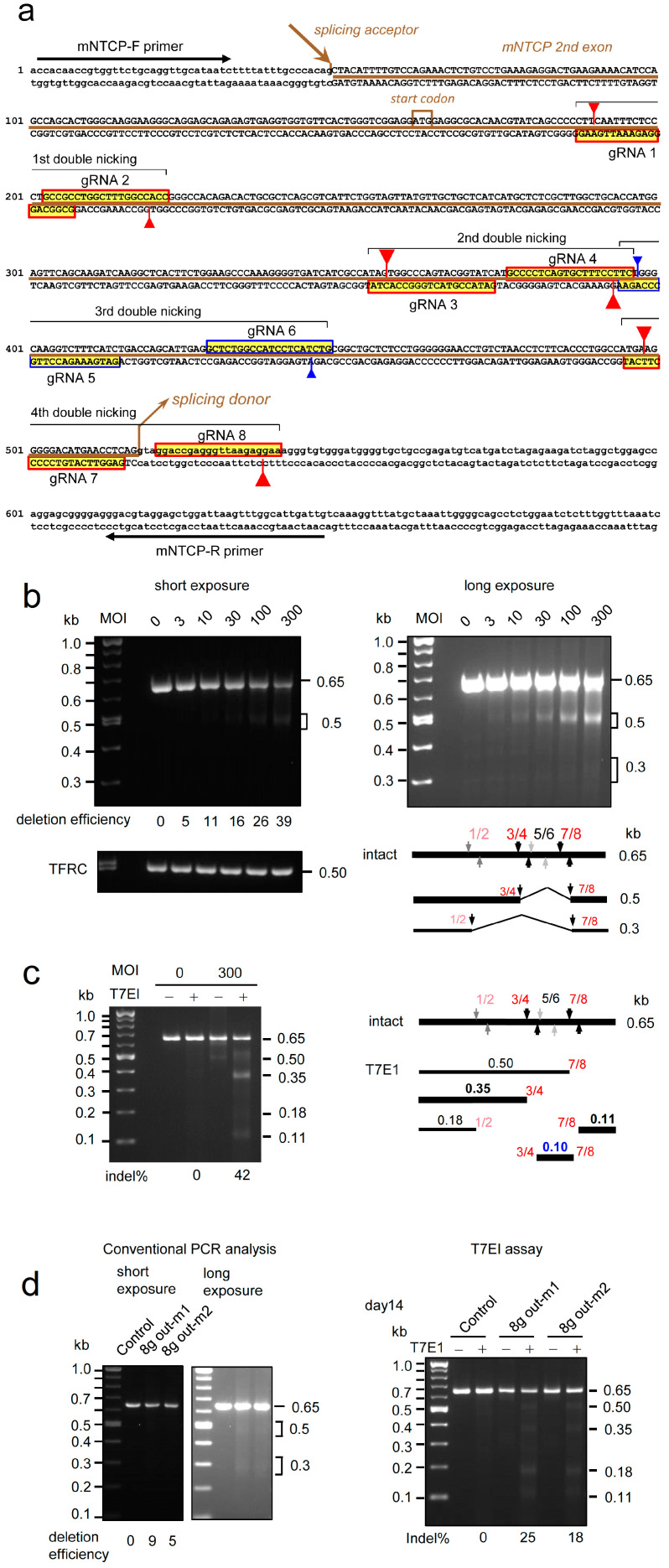
Genome editing of the mouse *NTCP* gene by all-in-one AdVs expressing eight gRNAs. (**a**) Targeted sequences of eight gRNAs expressed from Axda8sg-mNTCP-oCBNC9. Sequences of the second exon of the *NTCP* gene are shown. Bold brown line, second exon. The 20 nt recognition sequences of gRNA 1 to gRNA 8 are boxed in red or blue (gRNA 5 and gRNA 6). Red and blue arrowheads show the nicking positions by Cas9n. Long black arrows indicate the PCR primers used in (**b**–**d**). (**b**) Genome editing of the *NTCP* gene in vitro. MEF cells were infected with Axda8sg-mNTCP-oCBNC9 at MOIs of 3, 10, 30, 100, and 300. Total cellular DNA was extracted 3 days postinfection and subjected to PCR amplification using the primers shown in (**a**). (**left**) The intensity of the original 0.65 bp DNA decreased depending on the MOIs, and the deletion efficiency at MOI of 300 was 39%. The PCR products of the genome-edited DNA smaller than the original 0.65 kb DNA were quantified, and the deletion efficiency was calculated as ratios against the total DNA in the same lane. Because the conventional PCR used here detects only deletions but not indels, the genome editing efficiency is higher than the deletion efficiency shown here. As a control, the mouse TFRC gene was also PCR-amplified. (**right**) Broad band corresponding to approximately 0.5 kb and a region of approximately 0.3 kb were observed under the original 0.65 kb band in the long exposure. A schematic explanation of the PCR fragments is shown in the lower right panel. The major band of 0.5 kb at high MOIs (bold line) can be produced by deletion between guide 3/guide 4 (3/4, hereafter) and 7/8. The 0.3 kb fragment can be produced by double cleavages of 1/2 and 7/8 at high MOIs and is shown as a thin line. DNA fragments derived from the cleavage by 5/6 were not detected, indicating low cleavage efficiency of 5/6. (**c**) Indel rates analyzed by T7EI assay of MEF cells infected with Axda8sg-mNTCP-oCBNC9 at MOIs of 0 and 300. (**left**) Broad bands of approximately 0.35 kb and 0.11 kb detected under the 0.65 kb original DNA at MOI of 300. (**right**) Schematic explanation of the DNA fragments detected by T7EI assay. The major bands of 0.35 kb and 0.11/0.10 kb at MOI of 300 (bold line) can be produced by cleavage of 3/4 and 7/8. The 0.50 kb and 0.18 kb bands can be produced by single cleavage of 7/8 and 1/2. These fragments are shown as thin lines. DNA fragments derived from the single cleavage of 5/6 were not detected, as in (**b**). (**d**) In vivo genome editing of neonate mouse liver cells. Axda8sg-mNTCP-oCBNC9 was administered to two neonate mice via the facial vein with an 8 × 10^8^ rVT dose (four mice administered a 2.7 × 10^9^ rVT dose died the next day). The liver genomic DNAs were isolated 2 weeks after administration and were subjected to conventional PCR analysis and T7EI assay. By PCR, 0.5 kb and 0.3 kb DNA, derived from double cleavage of 3/4 and 7/8, were detected under the 0.65 kb original DNA, as in the in vitro analysis. By T7EI assay, the 0.50, 0.35, 0.18, and 0.11 kb DNA fragments were also produced, as shown in (**c**). These results indicate the high cleavage efficiency of 3/4 and 7/8 in vivo. The control consisted of DNA from untreated C57BL/6J mice.

## Data Availability

The data that support the findings of this study are available from the corresponding author upon reasonable request.

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
