# Peer review of "Construction and Stability of All-in-One Adenovirus Vectors Simultaneously Expressing Four and Eight Multiplex Guide RNAs and Cas9 Nickase"

_ijms, 2024, doi:10.3390/ijms25168783_

Round 1

Reviewer 1 Report

Comments and Suggestions for Authors

Co-infection method for efficient genome editing need co-introduction AAVs and AdVs, leading to an exponentially decreasing of the genome editing efficiency under some dilution conditions.  Here, authors constructed 4g all-in-one stable AdVs, achieving genome editing. Although the editing efficiency is low, authors discussed the possible reasons for this. The manuscript is well organized, and I recommend publishing it after addressing the problems.

Major comment:

1. I think it is necessary to compare the strength of original and shorten U6 promoters using a reporter gene.

2. The results of the genome editing efficiency of the all-in-one system is lower than those of the co-infection methods. Why did authors choose a high MOIs 300 to do this experiment? Did authors use the conditions they described in the “Introduction (line 123-125)” where the vectors are diluted, such as in vivo? Also, just compare the efficiency from their previous published paper is not enough, genome editing should be performed with both coinfection and All-in-one infection simultaneously, and the editing efficiency should be measured and compared.

If all the results cannot demonstrate the advantages of all-in-one vectors for genome editing, I think they need to show their unpublished data (described in line 534) and added this results in this manuscript.

Minor comments:

  1. The title is a little wordy, please shorten it.
  2. Line 82, E. coli should be Escherichia coli.
  3. Figure 2, 136bp and 20nt should be 136 bp and 20 nt.
  4. MOI or MOIs, authors should check the full text.
  5. References

Authors should check all the references, please unify them, some titles are capitalized (line 673, line 693 etc.) Some words should be changed to italics (such as in vivo, Staphylococcus aureus, etc.)

Reviewer 2 Report

Comments and Suggestions for Authors

The authors describe here the construction of adenovirus vector with four and eight gRNAs along with Cas9 nickase. There are a number of concerns that should be addressed by the authors,

Major concerns

1. The authors focus on reducing the size of the vector by generating the deletions to meet the packaging capacity but however the characterization of the effect of these deletions has not been performed carefully.

2. In figure 2, the authors show the deletion of a region in U6 promoter as a possible method for reducing the construct size. How does this deletion affect the promoter activity? This must have been studied further before proceeding. 

3. Based on figure 3, it is clear that AxdA double nicking efficiency was lower than that of AxdB. However further experiments were done with AxdA just because of the smaller size. As the authors mention in the discussion section, this could explain the lower gene editing efficiencies when compared to co-infection with two AdVs. 

4. The authors use conventional PCR and gel quantification or T7E1 assay to quantify the gene editing efficiency. To characterize the editing pattern better, it is advisable to perform Sanger sequencing, followed by ICE or TIDE analyses as these tools can calculate the frequency of different sizes of large deletion and other indels.

5. In lines 534 and 535, the authors have mentioned that the authors have found an alternate method for constructing the vector without compromising the editing efficiency but that the results will be published elsewhere. This raises the question on the relevance of the results described in this study when the authors have already found a better strategy for this vector construction.

Minor concerns

1. The authors should refrain from using the format of "figure x shows" (line 168 and 399) in the results section. They should instead describe the results and cite the figure numbers in brackets as per standard format. 

2. In figure 6 and 7, the authors should include the expected band sizes for the deletions that would be generated due to the deletions that can happen due to nicking with the different gRNAs. It would be easier for the readers to follow the description in the results section if there is a corresponding figure. 

Comments on the Quality of English Language

English language quality is fine. No major concerns to address.

Round 2

Reviewer 2 Report

Comments and Suggestions for Authors

The authors have addressed most of the concerns raised in the first round of review. However, it is not appropriate to exclude data which addresses the lower gene editing efficiency in the current study (lined 569-570) without providing a citation for the follow up study at least as part of a preprint. Modifying the backbone to address the low gene editing efficiency is very relevant to this study and so excluding this data is not very appropriate.

Comments on the Quality of English Language

English language quality is fine. 
